# Biomonitoring of Indoor Air Fungal or Chemical Toxins with *Caenorhabditis elegans* nematodes

**DOI:** 10.3390/pathogens12020161

**Published:** 2023-01-19

**Authors:** Sari Paavanen-Huhtala, Karunambigai Kalichamy, Anna-Mari Pessi, Sirkku Häkkilä, Annika Saarto, Marja Tuomela, Maria A. Andersson, Päivi J. Koskinen

**Affiliations:** 1Department of Biology, University of Turku, FI-20500 Turku, Finland; 2Aerobiology Unit, Biodiversity Unit of the University of Turku, FI-20500 Turku, Finland; 3Co-op Bionautit, Helsinki, FI-00790 Helsinki, Finland; 4Department of Microbiology, University of Helsinki, FI-00790 Helsinki, Finland; 5Department of Civil Engineering, School of Engineering, Aalto University, FI-02150 Espoo, Finland

**Keywords:** *C. elegans*, toxins, microbes, fungi, chemicals, biomonitoring

## Abstract

Bad indoor air quality due to toxins and other impurities can have a negative impact on human well-being, working capacity and health. Therefore, reliable methods to monitor the health risks associated with exposure to hazardous indoor air agents are needed. Here, we have used transgenic *Caenorhabditis elegans* nematode strains carrying stress-responsive fluorescent reporters and evaluated their ability to sense fungal or chemical toxins, especially those that are present in moisture-damaged buildings. Liquid-based or airborne exposure of nematodes to mycotoxins, chemical agents or damaged building materials reproducibly resulted in time- and dose-dependent fluorescent responses, which could be quantitated by either microscopy or spectrometry. Thus, the *C. elegans* nematodes present an easy, ethically acceptable and comprehensive in vivo model system to monitor the response of multicellular organisms to indoor air toxicity.

## 1. Introduction

Good indoor air quality is known to be crucial for human well-being, while airborne exposure to bioreactive, toxic or allergenic impurities compromises working capacity and health [1,2,3,4,5]. Indoor air problems may be due to technical problems such as malfunctioning ventilation, microbial impurities such as fungi associated with moisture damage [6,7,8,9,10,11,12,13] or chemicals enriched in indoor air and dust [13,14,15,16,17,18,19,20]. For example, some washing agents contain terpenoid-type odorants, such as limonene, which can react with indoor air ozone or oxidizing agents of disinfectants and form formaldehyde and other toxic carbonyl compounds such as glyoxal and methylglyoxal [21,22]. These compounds can be spread into the air as secondary organic aerosols (SOA) and thereby enter the human airways. In addition, under moisten and alkaline conditions, deterioration of plastic carpets and especially of their phthalate-based softeners may result in release of harmful volatile compounds, such as 2-ethyl-1-hexanol (2E1H) [23]. Thus, the symptoms observed in exposed individuals can be caused by synergistic or additive effects of multiple exposure agents. When the exposure time increases, acute symptoms may become chronic and lead to asthma or other diseases.

Even though multiple indicators for individual indoor air contaminants have already been developed, the total level of exposure may be more relevant for the health of an exposed person than the identity of the exposure agent(s). The risks of toxicity can be directly evidenced by in vivo experiments, in which laboratory animals are exposed to known concentrations of toxins in closed test chambers. However, such experiments are laborious and expensive, have ethical issues and may not even mimic the combined effects of various exposure agents that are associated with bad indoor air quality [24,25]. 

Some in vitro and ex vivo methods have previously been developed to measure the toxic or immunoreactive responses of bacteria (*E. coli*) or eukaryotic cells, such as porcine or hamster kidney cells, human macrophages, pulmonary epithelial cells or boar sperm cells. These methods have detected differences in intrinsic toxicity and immunoreactivity to indoor dusts or microbes isolated from building materials [26,27,28,29,30,31]. However, the causal connections between measured toxicity and human morbidity are still controversial [32,33]. Therefore, the described bioassays are not widely accepted for monitoring hazardous indoor air exposures or for distinguishing moisture-damaged buildings colonized by actively growing microbes from buildings without such problems [31]. In addition, single cell-based tests are not expected to fully recapitulate the effects of toxic agents on humans. Therefore, additional ethically acceptable in vivo biomonitoring methods are needed to evaluate effects of indoor air agents on multicellular organisms. 

The nematode *Caenorhabditis elegans* is a nonparasitic invertebrate that is widely used as a model organism and that has also been found to be suitable for biomonitoring of multiple types of harmful environmental agents [34]. Unlike vertebrates, *C. elegans* has neither a respiratory system nor a circulatory system, but relevant to this study, its chemosensory system can efficiently detect volatile odorants or soluble flavours [35,36]. Thereby, the nematode can discriminate between beneficial and harmful substances in its living environment, find food and avoid pathogens. The development of fertilized eggs to adult fertile nematodes takes only 3.5 days, so large amounts of them can be propagated under laboratory conditions in a fast and cost-efficient manner. Multiple types of *C. elegans* mutant strains are available as also transgenic reporter strains. Such strains expressing the green fluorescent protein (GFP) under the control of stress-responsive promoters have been shown to be useful in biomonitoring of environmental contaminants, such as heavy metals or organic pesticides [37,38]. For monitoring of indoor air problems, they have not previously been used, although they may also be suitable for this purpose. 

In this study, we evaluated the ability of transgenic *C. elegans* reporter strains to sense fungal or chemical toxins from samples collected from buildings with reported moisture damage or other indoor air problems. Pure toxins and their solvents were used as positive and negative controls, respectively, and the responses of the exposed nematodes were quantitatively monitored by microscopy and spectrometry. Our results provide preliminary evidence that *C. elegans* could be used as an indicator of indoor air safety.

## 2. Results

### 2.1. Stress-Responsive C. elegans Reporter Strains Sense Fungal Toxins

As transcriptional fusion reporters expressing GFP under the control of stress-responsive promoters in transgenic *C. elegans* strains had previously been shown to react to several types of environmental contaminants [37,38], we wanted to evaluate their ability to sense fungi and fungal toxins associated with indoor air problems. For this purpose, we isolated fungi from moisture-damaged buildings, cultured them on agar plates and suspended them in water or ethanol at different temperatures (20 °C, 50 °C or 100 °C) to obtain solutions containing either water- or fat-soluble toxins. While ethanol killed the fungi at all temperatures, as did water at higher temperatures, most fungus-derived toxins were expected to retain their efficacy.

In our initial experiments, we used synchronized day one adults from the *C. elegans sod4*::GFP strain (BC20333) that expresses GFP under the control of the *sod-4* promoter. The *sod-4* gene encodes superoxide dismutase and is expressed in the intestine, where its expression is enhanced in response to oxidative stress. The transgenic nematodes were first exposed in liquid culture to diluted (1:100) suspensions of *Stachybotrys sp*. (strain a/467) or to their solvents (water or ethanol). The fluorescence generated by the *sod4*::GFP reporter was observed by microscopy and quantitatively followed with spectrometry for up to 15 h. When we analysed nematodes exposed to water suspensions of *Stachybotrys sp*., we did not see any major differences as compared to solvent-treated controls (Figure 1A). By contrast, the fluorescence increased dramatically within a few hours in nematodes exposed to any of the ethanol suspensions. These results indicated that *C. elegans* responds to fat-soluble toxic compounds present in the *Stachybotrys* sp. samples. Toward the end of the experiment, absorbance of fluorescence was increased also in control samples, most likely due to the stressful effects of incubating *C. elegans* in high density in liquid culture in the absence of food.

When transgenic nematodes were exposed to ethanol suspensions prepared at 50 °C from five different *Stachybotrys* sp. strains, all of them induced fluorescent stress responses, which reached their maximal levels within 5 to 6 h (Figure 1B). However, there were differences in the strength of the responses, ranging from two- to four-fold increase in the fluorescence intensity as compared to the solvent control samples. For quantitation, the data from spectrometric tests were compared with each other by calculating the relative fold increases of absorbances from the start to the end of experiments.

Most fungus-derived toxins are highly resistant to heat [39,40], which explains why no major differences were observed between *Stachybotrys* sp. (a/467) samples incubated at 20°, 50° or 100 °C (Figure 1A and Figure 2A). By contrast, *Chaetomium* sp. (a/459) samples incubated at 50 °C caused stronger responses than those heated at 100 °C (Figure 2A), which can be explained by the fact that chaetoglobosins, the main toxins produced by *Chaetomium* species, are heat-sensitive [41]. Here, we also used another transgenic *C. elegans* strain *cyp-34A9*::GFP (BC20306). The *cyp-34A9* gene encodes a cytochrome P450 family member, which is expressed in the intestine, involved in metabolic processes and activated in response to xenobiotic stress. As fairly similar responses were obtained by both *cyp-34A9*::GFP and *sod-4*::GFP reporter strains, one or both of them were used in the rest of the experiments.

When transgenic nematodes were exposed to diluted (1:100) ethanol suspensions of six fungal samples isolated from moisture-damaged buildings, all of them induced two- to four-fold increases in fluorescent stress responses as compared to control samples (Figure 2B) or samples exposed to a nontoxic *Geotrichum candidum* (a/548) suspension. However, for the toxic samples, there were some differences in the kinetics of the responses. During the 24 h follow-up period, the fluorescence intensity continued to increase in the presence of all samples except for those containing *Stachybotrys* sp., the maximal response for which was obtained within 12 h. Microscopic examination of the samples revealed that while the fluorescent response to *Stachybotrys* exposure initially rose more rapidly than in the other samples, it reached its maximal level much earlier due to the death of the nematodes. Thus, the *Stachybotrys* suspension was lethal for them, while the strong responses induced by *Paecilomyces variotii* and *Trichoderma* sp. and the slightly milder responses by distinct *Aspergillus* species all remained sublethal at the concentrations we used. 

### 2.2. C. elegans Responses Correlate with Those Obtained from Single Cell-Based Assays

To compare our method with other recently introduced cell-based methods to evaluate indoor air toxicity, we exposed the *C. elegans* reporter strains to previously prepared ethanol extracts from fungal colonies that had been cultivated from dust samples collected from moisture-damaged buildings. Many of them had been shown to produce multiple mycotoxins that were toxic for either cultured cells (e.g., PK-15 kidney tubular epithelial cells) or boar sperm cells [42,43]. The biological activities of these mycotoxins have been listed in Appendix A. In addition, we analysed the effects of exudates that are liquid guttation droplets emitted by some fungal species [44]. Such exudates that have been detected near or on top of hyphae may contain even higher concentrations of mycotoxins than the hyphal biomass itself [45]. 

After spectrometric quantitation of *C. elegans* responses to various types of fungal extracts, our data were compared with those obtained from single cell-based tests. This comparison allowed us to classify the toxicity of distinct exposure agents based on their ability to induce strong (+++), moderately strong (++), mild (+) or no (-) responses in *C. elegans* (Table 1). As is evident from Table 2, all three tests were equally able to distinguish between the nontoxic *Penicillium* sample (TR) and its highly toxic (RcP61) relative. Most other toxin-producing samples were also well recognized by all the tests, except for the three *Trichoderma* samples (SJ40, THG and NJ14), which only produced strong responses in sperm cells. However, as *C. elegans* had reacted to another *Trichoderma* sp. sample (a/465) (Figure 2B), the differential results may reflect true differences between the toxin contents of the samples. 

To further test our method, we exposed the *C. elegans* nematodes in a blinded fashion to biomass dispersals of several indoor fungi that had previously provided positive or negative responses in single cell-based tests. Again, fairly similar results were obtained from the three types of tests (Table 3, Appendix A). Importantly, all the samples that had been tested negative in both of the single cell-based tests remained negative also in the *C. elegans* test. By contrast, positive responses were obtained for most, although not all, samples representing species associated with moisture damage (*Acremonium*, *Aspergillus*, *Exophiala* and *Paecilomyces*). However, here it should be noted that even these fungi do not always produce toxins, as toxin production depends on growth conditions.

To determine whether the *C. elegans* responses were dependent on mycotoxin dosages, we analysed some samples in more detail. When we exposed young adults to increasing concentrations of exudate fluid collected from the cultured indoor isolate of *Penicillium expansum* (RcP61), which had previously been shown to produce communesins A, B and D and chaetoglobosin C [45], we observed up to a two-fold increase in fluorescence with relatively small toxin concentrations (25–30 µg/mL; Figure 3A). The autofluorescence of the liquid was taken into account in the spectrometric measurements, but it did not interfere with the microscopic examination of the samples. Spectrometric quantitation of the effects of some extracts (e.g., ABCD) could have been compromised due to the strong autofluorescence of the exposure agent, while for nonfluorescent samples such as OT7, we could obtain a clear dose-dependent response based on the concentration of chaetomin (17–33 µg/mL; Figure 3B). As fungal extracts always consist of heterogenous mixtures of both toxic and nontoxic compounds, some mycotoxins were tested as pure solutions. For example, emodin produced by *Aspergillus* species caused fluorescent responses at concentrations as low as 2.5 µg/mL. 

### 2.3. Indoor Chemicals Reduce C. elegans Motility in a Dose-Dependent Fashion

Indoor air quality may also be reduced by nonmicrobial contaminants that are derived, e.g., from cleaning agents or their oxidized derivatives. To represent these, we analysed the responses of *C. elegans* to glyoxal and methylglyoxal. For this purpose, the nematodes were exposed to increasing concentrations of them in liquid cultures for 24 h. In addition, nematode motility was imaged and analysed with software that was recently developed to measure movements of sperm cells in response to environmental factors [43]. According to these analyses, both glyoxal and methylglyoxal were highly toxic to *C. elegans* (Figure 4A). However, methylglyoxal was about five times more toxic than glyoxal, as a 0.1% solution of methylglyoxal already reduced the motility of nematodes, and a 1% solution killed them (Figure 4B). There was very little variation between the video recordings from similarly exposed nematode populations, indicating that reproducible data on motility can be obtained with this method.

To compare the efficacy of *C. elegans* and cell-based assays also in biomonitoring of chemicals, we analysed responses of the two *C. elegans* reporter strains to five chemical surfactants that are used as detergents in cleaning chemicals and hygiene products. These included one anionic detergent (SDS), one cationic detergent (DDDAC) and three nonionic detergents (Tween 80, Triton X-100, Genapol X-080). Results on *C. elegans* responses, shown in Appendix A, were then compared with previously reported results from cell proliferation and sperm motility tests [46]. Based on conclusions summarized in Table 4, DDDAC was highly toxic in all tests, while Genapol-X-080 and Triton X-100 were not cytotoxic in the cell proliferation assay, but they caused moderate or strong fluorescent responses in both *C. elegans* reporter strains, respectively. Both Genapol-X-080 and Triton X-100 also reduced motility of sperm cells but had milder effects on *C. elegans* motility. Tween 80, in turn, was not toxic to *C. elegans* or cells but moderately reduced motility of both *C. elegans* and sperm cells, while SDS only affected sperm motility. The dose-dependent effects of the distinct surfactants on the fluorescent responses and motility of the reporter strains were also evident in the microscopic images taken from them (Figure 5).

### 2.4. C. elegans Reporter Strains Also Respond to Airborne Exposure to Fungi

As many fungi produce spores or volatile compounds that are released to the air, we wanted to test the suitability of the transgenic *C. elegans* nematodes for monitoring airborne compounds. Therefore, we incubated young adults together with their food bacteria on two-compartment petri dishes, where they were subjected via airborne exposure to fungi that were growing on the other side with their own growth medium. These exposures resulted in milder and more variable fluorescent responses than those observed in liquid assays with fungal suspensions. For example, the *Stachybotrys* a/467 strain, which produced even four-fold responses over controls in liquid assays (Figure 1), now generated slightly milder but still remarkable responses that could be quantitated by both microscopy and spectrometry (Figure 6A,B). Additional examples are shown in Figure 6C and listed in Table 5, according to which the transgenic nematodes responded to several different fungal strains but not to the nontoxic *Geotrichum candidum*. 

In addition to fungal colonies, we tested airborne responses of *C. elegans* to clean or moisture-damaged carpets by placing pieces of them to the other side of the two compartment petri dishes. In this case, we observed stronger responses against damaged carpets as compared to controls (Figure 7A), as quantitated by microscopy. We then incubated four moisture-damaged carpet samples in air-tight glass desiccators for three days and analysed the volatile organic compounds (VOCs) produced by them. When the total amounts of VOCs (TVOCs) were determined as toluene equivalents (Table 6), their concentrations in three out of four cases exceeded the action limit value determined by the Finnish Institute for Occupational Health for healthy indoor air (400 µg/m^3^) [47]. A more detailed analysis of the composition of the VOCs revealed that most of them represented 2-ethyl-1-hexanol (2E1H), which in all four cases remarkably exceeded its action limit value (10 µg/m^3^) [47]. C9 and C10 aliphatic alcohols were also found in these samples as well as smaller amounts of other compounds. To further investigate the effects of pure 2E1H on *C. elegans*, we carried out chemotactic analyses, where the movement of animals either towards or away from different concentrations of the compound were measured. As shown in Figure 7B, 2E1H was highly aversive to both of our *C. elegans* strains, among which differences were not expected, as the transgenes should not affect the olfactory sensations of the animals. For comparison, we carried out chemotaxis assays with 1-octen-3-ol, which is produced by several types of fungi, which was expected to be as aversive for *C. elegans* as octanol [35] and which efficiently repelled the nematodes in our experiments in a dose-dependent fashion (Figure 7C). 

## 3. Discussion

In the present study, we show that suspensions, extracts or exudates of mycotoxin-producing fungi induce clear stress responses in transgenic *C. elegans* strains expressing fluorescent stress-responsive reporters. For this purpose, we used two distinct reporter strains, which expressed GFP under the control of promoters responsive to either oxidative (*sod-4*) or xenobiotic (*cyp-34A9*) stress. We had previously tested several other reporter strains, but these two were chosen, as they gave the most promising and most reproducible results, which usually supported each other. Some fluorescence for them could also be observed under control conditions, but the fluorescence intensity was significantly increased upon exposure to toxic samples, as confirmed by statistical analyses.

Based on our data, the fluorescent responses can be quantitated and classified by both spectrometry and microscopy, the results of which support each other. While spectrometric plate readers are faster and easier to use for quantitation, microscopic images also provide qualitative information, e.g., on the localization of the responses. By imaging, we can also measure fertility, motility and mortality of the nematodes and to distinguish between lethal and sublethal responses. Moreover, fluorescence produced by the exposure agents themselves does not as easily interfere with microscopic analyses as compared to spectrometry, where the autofluorescence needs to be taken into account during quantitation. 

Our data indicate that already within 24 h of exposure, the *C. elegans* nematodes develop prominent fluorescent responses to such fungi or their metabolites that have also been shown to be toxic in single cell-based tests, while no significant responses were observed to nontoxic control samples. Thus, with our methodology, we can distinguish potentially health hazardous agents from harmless ones. Based on quantitation of the response data, the exposure agents can be classified into strongly toxic (>2x response), moderately toxic (1.5–2x), mildly or variably toxic (1.25–1.5x) and nontoxic (<1.25x). However, there are differences in the effectiveness of distinct types of agents. 

No effects were observed upon exposure to extracts of harmless fungi such as *Geotrichum candidum*, which is known to be common in soil but also part of the normal human microbiome. By contrast, strong responses were obtained by various toxic extracts, including those prepared from colonies of *Stachybotrys* sp., *Aspergillus* species and *Paecilomyces variotii*, the abundant presence of which has been associated with moisture damage and indoor air problems [48,49]. By contrast, *Penicillium* family members are fairly common in all types of buildings, and only some of them produce toxins [49,50], explaining the variable responses we observed. In addition, it should be noted that mycotoxin production depends on the metabolic activity and growth phase of fungi, so not even the pathogenic species produce toxins all the time, but only under certain conditions [51]. 

*Stachybotrys* extracts were even lethal for *C. elegans*, while the others caused sublethal responses with the up to 5% (*w*/*v*) concentrations used. Ethanol suspensions had more striking effects than water suspensions, which correlated well with the previous observations, according to which hydrophobic lipid-soluble toxins extracted by ethanol are more hazardous than water-soluble hydrophilic ones, as they can penetrate cellular membranes [52]. Exudates produced by certain fungi (*Aspergillus versicolor* and *Penicillium expansum*) were also highly toxic. These exudates containing lipid-soluble toxins can be suspended in the air and thereby enter the airways and cause health issues there [31].

The extracts prepared from fungi that had been cultivated from dust or other materials collected from moisture-damaged buildings can inform us about the potential toxicity of the observed species but not whether they have truly produced toxins in the point of collection. To measure acute toxicity, we exposed the *C. elegans* nematodes to fungal cultures that were grown on the other side of a two-compartment petri dish. This kind of a setting had successfully been used by others to study the pathogenic effects of some microbial volatiles [53]. However, the responses following airborne exposures were slightly milder or more variable than those of liquid assays, most likely reflecting the ability of the fungi to disperse toxic spores or volatiles into the air. Here, it should be noted that the toxicity may depend on the origin of the microbe and the duration that it has been grown as a pure culture under favourable conditions. When the microbe does not have to compete with or repel other species, it may not produce the same quantity of toxins as under more unfavourable environmental conditions containing multiple species. Moreover, the effectivity of toxins largely depends on their route of exposure. Once inhaled, the same substance can be much more hazardous for humans than when taken orally, so even modest airborne responses can be significant.

Airborne exposures to moisture-damaged materials such as plastic carpets also induced clear fluorescent responses in *C. elegans*, which correlated well with the presence of various harmful VOCs, the concentrations of which exceeded the action limit values for healthy air. Interestingly, most of the identified VOCs represented 2-ethyl-1-hexanol (2E1H), the amounts of which exceeded both the Finnish (10 µg/m^3^) [47] and the EU LCI (lowest concentrations of interest, 300 µg/m^3^) [54] limits. 2E1H can be produced by several microbes, such as *Aspergillus versicolor*, and, together with C9 alcohols, it has been used as an indicator for degradation of phthalates (softeners of plastic carpets) under moist and alkaline conditions [23]. The fat-soluble 2E1H that can accumulate in the human body under prolonged exposure was also shown to be aversive to *C. elegans* nematodes in chemotaxis assays, indicating that they can smell these types of substances and recognize them as unpleasant for themselves.

Other types of chemicals can also compromise the indoor air quality. The strong fluorescent responses obtained with diluted methylglyoxal and glyoxal solutions are of interest, as in the presence of ozone or oxidizing disinfectants, methylglyoxal and glyoxal can be produced to the air from odorants, such as limonene, which are often present in cleaning agents used in public buildings [21,55]. This occurs especially easily when using the so-called leave-on processes, where cleaning agents are not washed away but are left on the surfaces, from where they can become evaporated. Oxidizing disinfectants can be present in the cleaning agents themselves, while ozone can be formed due to the use of copy machines or air purifiers. If glyoxal and methylglyoxal are lethal for *C. elegans*, it is highly likely that they are also hazardous for humans. The crippling effects of these compounds were also manifested through the decreased motility of the nematodes, which could be quantitated from videos. Many cleaning agents also contain surfactants, which lower the surface tension of liquids and thereby help to spread them. Some of them, including the cationic surfactant DDDAC and the nonionic surfactants Genapol X-080 and Triton X-100, caused strong fluorescent responses in *C. elegans* and reduced their motility. Interestingly, the exudates produced by actively growing indoor air fungi can also contain water-repelling surfactants [56]. 

Using shared samples, our method was compared in a partially blinded fashion against two single cell-based tests that measured reduction of the proliferation of cultured cells or motility of boar sperm cells [42,43]. While there were some differences in the sensitivity of the three bioassays towards distinct types of fungal samples, most of the toxic samples were recognised by all of them. Importantly, the *C. elegans* test did not give positive results to nontoxic control samples or samples that had given negative results by the two other tests. Furthermore, as a multicellular organism, *C. elegans* may provide additional advantages over the tests with single types of cells. Unlike cells, *C. elegans* can both taste and smell compounds, and may therefore make more comprehensive conclusions on whether certain compounds or their combinations are toxic, irritating or otherwise harmful. Furthermore, as it can recognise and respond to volatiles produced in real time by microbes or moisture-damaged materials, it may provide more rapid and more accurate data on the presence of acute toxicity than cellular assays, for which microbial samples first need to be cultivated on plates for a week or two. After such cultivation, the enriched species may only partially represent the flora of the original sampling place or may not produce toxins in a similar fashion. Furthermore, by using microscopy with distinct *C. elegans* reporter strains, it may be possible to distinguish general stress responses from tissue-specific ones in the respiratory, nervous or immune systems, providing a clear advantage over assays based on the responses of single types of cells. Yet the tests based on mammalian cells may help to confirm that samples observed to be toxic for *C. elegans* are hazardous also for mammals. Thus, for diagnostic purposes, a combination of different types of tests and techniques is expected to be more informative than single types of approaches alone.

## 4. Conclusions

Taken together, our data promisingly show that exposures to agents that are toxic to mammalian cells can also cause clear stress responses in the transgenic *C. elegans* reporter strains. In addition to fungi and their toxins, buildings with indoor air problems also contain other harmful chemicals, which can similarly be detected with the help of nematodes. Thus, the *C. elegans* nematodes offer a versatile, unbiased and comprehensive method to monitor overall indoor air quality. They cannot inform us of what kind of compounds there are in the air, but they may give preliminary information on the risks of exposure to poor quality indoor air and on the needs for more thorough technical investigations. Thus, further studies are warranted to develop not only laboratory tests with *C. elegans* but also additional applications for field tests.

## 5. Materials and methods

### 5.1. Experimental Design

The distinct types of samples and methods used in this study have been summarized in Figure 8.

### 5.2. C. elegans Nematode Strains and Their Maintenance

The transgenic *C. elegans* strains BC20333 (*sod-4*::GFP) and BC20306 (*cyp-34A9*::GFP) have been previously described [37] and were kindly provided by David De Pomerai (University of Nottingham, UK). These integrated reporter strains express the green fluorescent protein (GFP) under the control of promoters that are responsive to either oxidative (*sod-4*) or xenobiotic (*cyp-34A9*) stress. 

The *C. elegans* strains were grown and maintained at 20 °C on NGM (nematode growth medium) agar plates seeded with the *E. coli* strain OP50 or its streptomycin-resistant derivative OP50-1 using standard culturing methods [57,58]. Before experimentation, the *C. elegans* populations were synchronized by bleaching. For liquid assays, well-fed day one adults were washed three times with K medium (53 mM NaCl, 32 mM KCl) to remove bacteria, after which aliquots of 800–900 animals in 295 µL of K medium were placed to wells of 96-well plates with optically transparent bottoms (Black IsoPlate-96, PerkinElmer, Waltham, MA, USA). Then the animals were exposed in triplicates to increasing concentrations of exposure agents. For airborne exposures, two-compartment petri dishes (Greiner Bio-One, Thermo Fisher Scientific, Waltham, MA, USA) were used, in which the *C. elegans* nematodes were grown on one side and the microbes or materials to be analysed on the other side, both on their own optimal growth media. 

### 5.3. Spectrometric Analyses

The intensity of the GFP fluorescence produced by the *C. elegans* samples was spectrometrically measured in the beginning and at the end of the 24 h liquid exposures (by the EnSpire plate reader, PerkinElmer, Turku, Finland) or every 30 min during the exposure (by the Hidex Sense plate reader, Hidex, Turku, Finland). Wells containing only K medium were used as blanks. The autofluorescence produced by some of the exposure agents was reduced from the fluorescence produced by the samples with exposed nematodes. After airborne exposures of one to three days, the nematodes grown on plates were washed three times with K medium to remove bacteria and then subjected to microscopy and/or spectrometry. 

### 5.4. Microscopic Analyses

Samples of control or exposed *C. elegans* nematodes were immobilized with 0.2 mM levamisole hydrochloride (Vetranal, Sigma–Aldrich), mounted on glass slides and imaged by light microscopy (Olympus CK40 or Leitz Fluovert FS) or fluorescence microscopy (Zeiss Axiovert M200). Nematodes on optical 96-well plates were also directly imaged using the Zoe imager (Bio-Rad Laboratories, Hercules, CA, USA). The intensity of fluorescence was measured from digital images using the ImageJ software (Wayne Rasband, NIH, USA). 

### 5.5. Motility Measurements 

To analyse the effects of exposure agents on the motility of exposed nematodes, short videos taken from liquid assay samples were analysed by a Matlab-based software, which had originally been developed to measure sperm motility [43]. The relative motility of nematodes was determined in comparison to unexposed populations (motility 100%) and completely immobilized populations (motility 0%). 

### 5.6. Chemotactic Assays

To determine whether volatile compounds were attractive or aversive to the *C. elegans* nematodes, chemotactic assays were performed as described previously [35,58]. Briefly, 50–100 well-fed and washed young adults were placed in the middle of an NMG agar plate without food bacteria. A drop of an odorant or its dilution was placed on one side of the plate, and a drop of its solvent, usually ethanol, on the other side. Drops of 1 M sodium azide had been added in advance to both spots in order to paralyze nematodes reaching them. After 30 to 60 min incubation, all the nematodes that had moved out from their original spot were counted from both sides, and chemotactic indices (CI = O − C/O + C) were calculated as the number of nematodes that had moved towards the odorant (O) minus the number of nematodes towards the control (C), divided by the total number of them. Attractive odorants were expected to give positive CI values, and aversive ones negative values. 

### 5.7. In Vitro and Ex Vivo Assays

The protocols for the in vitro assay to measure proliferation of porcine tubular kidney (PK-15) or baby hamster kidney (BHK-21) cells and the ex vivo assay to measure motility of boar sperm cells were performed as previously described [43,59]. 

### 5.8. Preparation of Extracts or Collection of Liquid Exudates from Fungal Pure Cultures

The indoor fungal strains listed in Table 7 were isolated from moisture-damaged building materials (except for the nontoxic *Geotrichium candidum* control strain) and cultivated at room temperature on 2% malt extract agar (MEA), pH 5.5 or dichloran-glycerol (DG18) plates for 7 to 10 days. Then 10 mg biomass (wet wt) from a culture plate was suspended in 0.2 mL of ethanol or water, and the vial was sealed and heated in a water bath for 10 min at 20 °C, 50 °C or 100 °C. The obtained suspensions were used to expose *C. elegans* nematodes in liquid-based assays as described above.

The cultivation of indoor fungal strains listed in Table 8, preparation of ethanol extracts from their biomasses and identification of mycotoxins produced by them have been published previously [42,45,59,60,61]. Briefly, fungi were cultivated at room temperature on MEA or DG18 plates for 7 to 10 days. The harvested fungal biomasses, 100–400 mg wet wt, were flooded with 100 mL of ethanol overnight. The following day, the soluble material was separated from biomass debris by centrifugation (1800× *g* for 10–20 min). Then the clear supernatants were allowed to evaporate to dryness at 50 °C, weighed and redissolved in 99% ethanol to concentrations of 10 mg dry solids per mL. The extracts were previously used in in vitro assays with porcine kidney tubular cells (PK-15) and ex vivo assays with boar sperm cells [43,59,60,61] and were now used in in vivo assays with *C. elegans*.

The collection of liquid exudates from fungal biomass has been described previously, as has analysis of the mycotoxin contents of the exudates produced by the *Penicillium expansum* strain RcP61 [45]. Briefly, fungal pure cultures inoculated on MEA plates were sealed with a gas-permeable tape and incubated for 5–10 days with the lid upwards. The cultures were inspected daily by the naked eye or under stereomicroscope with UV light. Droplets were collected from the inner surface of the Petri dish lid or the surface of the fungal biomass, diluted into an equal volume of ethanol and heated at 50 °C for 10 min. The exudates were used for toxicity testing in the three bioassays similarly as the extracts.

Biomass dispersals of indoor fungal strains listed in Table 9 were prepared of isolates cultivated on MEA or DG18 plates. The strains were identified to genus/species level by classical morphological methods [62] of plate-grown fungal colonies. Then 10–20 mg biomass (wet wt) from a culture plate was looped into 0.2 mL of ethanol, and the vial was sealed and heated in a water bath for 10 min at 50 °C. The obtained ethanolic biomass dispersal was used to expose boar sperm or BHK-21 cells. The lysate was considered toxic when 2.5 vol% of the extract inhibited sperm motility within 30 min or 1 day or 5 vol% of it inhibited target cell proliferation within 2 days [45]. The biomass dispersals were used for toxicity testing in the three bioassays similarly as the extracts and exudates. However, for *C. elegans*, the tests were carried out in a blinded fashion, so that the identity of the samples was revealed only afterwards to allow for reliable comparison with the previously obtained data from the two single cell-based assays, which had been carried out by Co-op Bionautit (Helsinki, Finland). 

### 5.9. Other Materials

Some building materials such as pieces of deteriorated plastic carpets were also analysed as such. Chemicals such as emodin, glyoxal, methylglyoxal, 2-ethyl-1-hexanol and 1-octen-3-ol were purchased from Sigma–Aldrich (St. Louis, MO, USA) and diluted into ethanol as indicated. The surfactants included the anionic detergent SDS (sodium dodecyl sulphate, Sigma–Aldrich), the cationic detergent DDDAC (didecyldimethylammonium chloride; Merck, Darmstadt, Germany) and three nonionic detergents, Genapol X-080 (polyethyleneglycol monoalkyl ether), Triton X-100 (polyethyleneglycol-p-is-octylphenyl ether) and Tween 80 (polyethylene glycol sorbitan monooleate), all from Sigma–Aldrich. DDDAC was diluted into methanol and other surfactants into water.

### 5.10. VOC Analyses

To analyse the amounts and nature of volatile compounds (VOCs) emitted by moisture-damaged plastic carpet samples, they were incubated in air-tight glass desiccators for three days. The VOC contents were then analysed by the Finnish Institute for Occupational Health (Helsinki, Finland). 

### 5.11. Statistical Analyses

The data and their statistical significance were analysed by the Excel software (Microsoft Office), which was also used to prepare bar and line charts as well as boxplot data. Statistically significant differences (*p* < 0.05) determined by the Student’s *t*-test were marked with asterisks. Error bars represent standard deviations.

## Figures and Tables

**Figure 1 pathogens-12-00161-f001:**
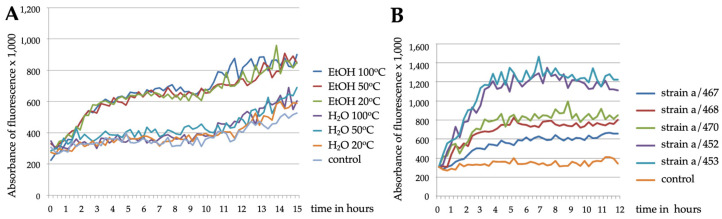
Stress-responsive reporter strains sense fungal toxicity. Fluorescent responses of the *sod-4*::GFP strain to ethanol or water suspensions of *Stachybotrys* sp. (a/467) (**A**) or to ethanol suspensions of indicated *Stachybotrys* sp. strains (**B**), as analysed by spectrometry.

**Figure 2 pathogens-12-00161-f002:**
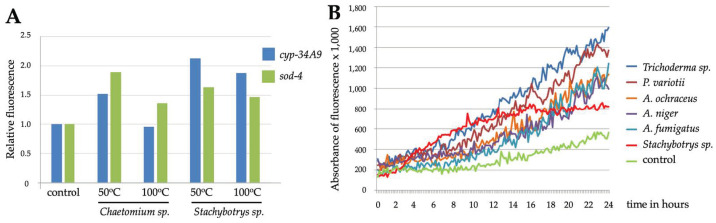
(**A**) Fluorescent responses of the *cyp-34A9*::GFP or *sod-4*::GFP strains to ethanol suspensions of *Chaetomium* sp. (a/459) or *Stachybotrys* sp. (a/467) prepared at 50 °C or 100 °C. (**B**) Fluorescent responses of the *cyp-34A9*::GFP strain to ethanol suspensions of several species of fungi: *Trichoderma* sp. (a/465), *Paecilomyces variotii* (a/462), *Aspergillus ochraceus* (Asp25), *Aspergillus niger* (a/464), *Aspergillus fumigatus* (a/466) and *Stachybotrys* sp. (a/468).

**Figure 3 pathogens-12-00161-f003:**
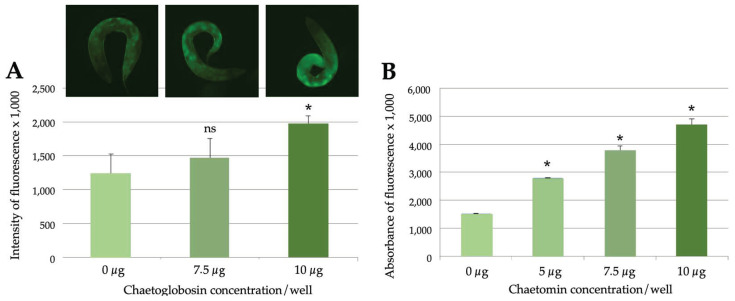
The *C. elegans* responses are dependent on mycotoxin dosage. (**A**) Effects of chaetoglobosin-containing *Penicillium expansum* (RcP61) exudates on fluorescent responses of the *sod-4*::GFP strain, as analysed by microscopy. Shown above are representative fluorescent images of individual nematodes. (**B**) Effects of chaetomin-containing samples (OT7) on fluorescent responses of the *cyp-34A9*::GFP strain, as analysed by spectrometry. Student’s *t*-test was used to analyse the statistical significance of the data. Significant differences (*p* < 0.05) as compared to control samples were marked with asterisks; ns refers to no significance. Error bars represent standard deviations.

**Figure 4 pathogens-12-00161-f004:**
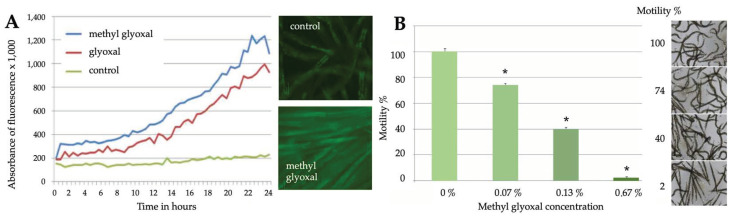
Chemical toxins cause fluorescent responses in *C. elegans*, but also reduce their motility. (**A**) Time-dependent effects of glyoxal (0.2%) and methylglyoxal (0.04%) on fluorescent responses of the *cyp-34A9*::GFP strain, as analysed by spectrometry. Shown on the right are representative fluorescent images of nematode populations. (**B**) Dose-dependent effects of methylglyoxal on the motility of the *sod-4*::GFP strain, as analysed by video recordings. Student’s *t*-test was used to analyse the statistical significance of the data. Significant differences (*p* < 0.05) as compared to control samples were marked with asterisks. Error bars represent standard deviations. Shown on the right are representative brightfield images of nematode populations.

**Figure 5 pathogens-12-00161-f005:**
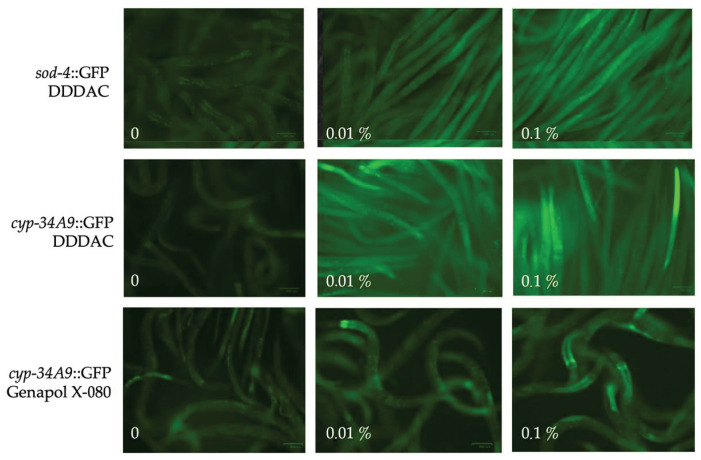
Representative images of the dose-dependent effects of DDDAC and Genapol X-080 on the fluorescent responses and motility of *sod-4*::GFP and *cyp-34A9*::GFP strains.

**Figure 6 pathogens-12-00161-f006:**
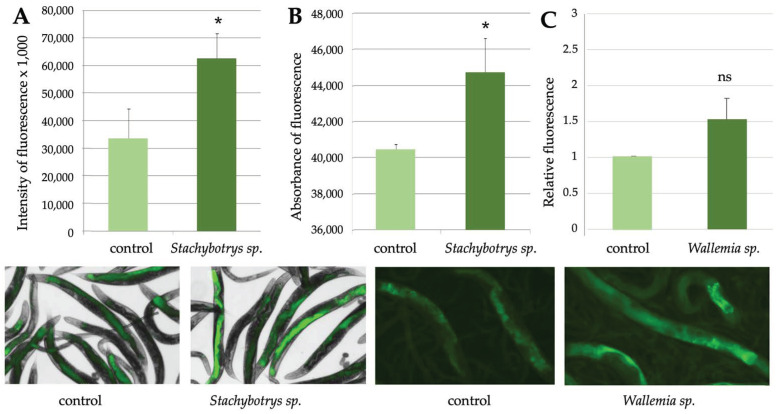
Airborne exposure to fungal cultures also causes fluorescent responses in *C. elegans.* Effects of *Stachybotrys* sp. exposure on fluorescent responses of the *cyp-34A9*::GFP strain, as analysed by microscopy (**A**) or spectrometry (**B**). (**C**) Relative effects of *Wallemia sp*. exposure on fluorescent responses of the *sod-4*::GFP strain, as analysed by microscopy. Student’s *t*-test was used to analyse the statistical significance of the data. Significant differences (*p* < 0.05) as compared to control samples were marked with asterisks; ns refers to no significance. Error bars represent standard deviations. Shown below are representative images of nematode populations. In those on the left, fluorescent images have been combined with brightfield images to highlight the localization of the fluorescence.

**Figure 7 pathogens-12-00161-f007:**
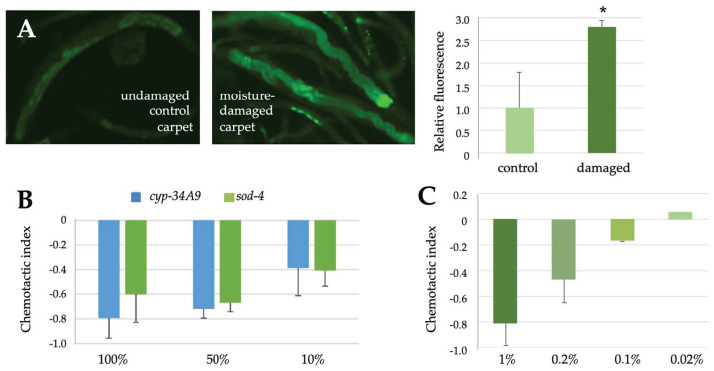
Moisture-damaged materials emit toxic volatile compounds. (**A**) Representative images of the fluorescent responses of the *cyp-34A9*::GFP strain to airborne exposure to control or moisture-damaged carpets, as analysed by microscopy, and the quantitation of them on the right. In chemotactic assays, the aversive effects of 2-ethyl-1-hexanol (**B**) or 1-octen-3-ol (**C**) on *C. elegans* were analysed using indicated dilutions in ethanol. Student’s *t*-test was used to analyse the statistical significance of the data. Significant differences (*p* < 0.05) as compared to control samples were marked with asterisks. Error bars represent standard deviations.

**Figure 8 pathogens-12-00161-f008:**
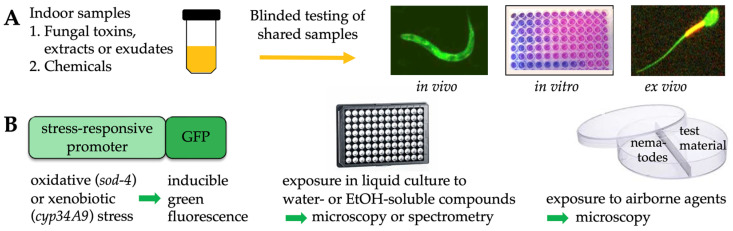
Experimental design. (**A**) Using shared fungal or chemical samples, the in vivo responses of the *C. elegans* nematodes were tested in a blinded fashion, and the results were compared to those previously obtained from in vitro PK-15 or BHK-21 cell proliferation assays and ex vivo sperm motility assays. (**B**) Our *C. elegans* bioassay is based on inducible green fluorescence produced upon liquid-based or airborne exposure to oxidative stress (with the *sod-4* promoter) or xenobiotic stress (with the *cyp-34A9* promoter).

**Table 1 pathogens-12-00161-t001:** Classification of toxic responses to exposure agents in in vivo, in vitro and ex vivo bioassays with *C. elegans* (*Ce*), PK-15 or BHK-21 cells or boar sperm, respectively.

Models/Toxicity of Exposure Agents	Strongly Toxic +++	Moderately Toxic ++	Mildly Toxic +	Nontoxic -
Relative increse in *Ce* fluorescence	>2 x	1.5–2 x	1.25–1.5 x	<1.25 x
Decrease in cell proliferation (EC_50_)	<5 µg/mL	5–15 µg/mL	15–50 µg/mL	>50 µg/mL
Decrease in sperm motility (EC_50_)	<5 µg/mL	5–10 µg/mL	10–25 µg/mL	>25 µg/mL

**Table 2 pathogens-12-00161-t002:** Effects and identified mycotoxins of fungal extracts ^1^ or exudates ^2^ in in vivo, in vitro and ex vivo bioassays with *C. elegans*, PK-15 cells and boar sperm cells, respectively. ND, not determined.

Strain Code	Species/Genus	Mycotoxins	*C. elegans*	Cells	Sperm
POB8	*Agrostalagmus* *luteoalbus* ^1^	melinacidins	+	++	++
K20	*Aspergillus* *versicolor* ^1^	sterigmatocystin	++	+++	++
SL3	*Aspergillus* *versicolor* ^2^	sterigmatocystin, averufin	+	+++	++
MH1	*Chaetomium* ^1^	chaetoglobosin	+++	+	++
OT7	*Chaetomium* *cochliodes* ^1^	chaetomin, chaetoviridins	++	+++	++
ABCD	*Chaetomium* *globosum* ^1^	chaetoglobosin, chaetoviridins	+++	+	++
MH5	*Chaetomium* *globosum* ^1^	chaetoglobosin, chaetoviridins	+++	+	++
TR	*Penicillium sp*. ^1^	nontoxic	-	-	-
RcP61	*Penicillium* *expansum* ^2^	chaetoglobosin, communesins	+++	+++	+++
SJ40	*Trichoderma* *citrinoviride* ^1^	trilongins	-	+	+++
THG	*Trichoderma* *longibrachiatum* ^1^	trilongins	+	+	+++
NJ14	*Trichoderma* *trixiae* ^2^	ND	+	++	+++

**Table 3 pathogens-12-00161-t003:** Effects of biomass dispersals of indoor fungi in in vivo, in vitro and ex vivo bioassays with *C. elegans* (*Ce*) strains, BHK-21 cells and boar sperm, respectively. ND, not determined.

Strain Code	Species/Genus	*Ce sod-4*	*Ce cyp-34A9*	Cells	Sperm
A21	*Acrenium* sp.	+	+	-	++
BA36	*Aspergillus* sp.	++	+++	++	+++
A4	*Aureobasidium pullulans*	++	++	++	+
A5	*Aureobasidium pullulans*	-	-	++	+
A6	*Aureobasidium pullulans*	-	ND	-	-
BA35	*Aureobasidium pullulans*	+	+	+	-
A3	*Exophiala* sp.	-	-	-/+	-
BA32	*Exophiala* sp.	++	ND	++	-
L6	*Paecilomyces* sp.	-	-	-	-
L21	*Paecilomyces* sp.	-	ND	-/+	-
BA37	*Rhizomucor* sp.	-	ND	-	-

**Table 4 pathogens-12-00161-t004:** Effects of chemical surfactants on *C. elegans* strains, PK-15 cells and boar sperm.

Chemical Compound	Detergent Type	*Ce sod-4* Fluorescence	*Ce cyp-34A9* Fluorescence	*Ce* Motility	Cell Proliferation	Sperm Motility
DDDAC	cationic	+++	++	+++	+++	+++
Genapol X-080	non-ionic	++	++	+	+	+++
Triton X-100	non-ionic	+++	+++	++	+	+++
Tween 80	non-ionic	-	-	++	-	++
SDS	anionic	-	-	-	-	++

**Table 5 pathogens-12-00161-t005:** Fluorescent responses of *C. elegans* reporter strains to airborne exposure to cultures of distinct fungal strains.

Strain Code	Species/Genus	*Ce* Fluorescence
a/464	*Aspergillus niger*	++
Asp25	*Aspergillus ochraeus*	+++
a/459	*Chaetomium* sp.	+
b/548	*Geotrichium candidum*	-
a/462	*Paecilomyces variotii*	+
a/467	*Stachybotrys* sp.	+++
a/465	*Trichoderma* sp.	+
a/456	*Wallemia* sp.	++

**Table 6 pathogens-12-00161-t006:** Volatile organic compounds (VOCs, µg/m^3^) emitted during three days by moisture-damaged carpet samples to four (1–4) air-tight glass desiccators. TVOCs, total amounts of VOCs determined as toluene equivalents; DMCPS, decamethylcyclopentasiloxane; PMH, 2,2,4,6,6-pentamethylheptane.

VOCs/Chamber	1	2	3	4
TVOCs	1500	160	560	550
1-butanol	19	3	-	6
2-ethyl-1-hexanol	830	240	460	770
C9/C10 alcohols	880	-	220	-
a-pinene	62	-	15	-
ß-pinene	13	-	-	-
DMPCS	6	-	-	11
toluene	-	3	-	5
PMH	-	-	17	-
2-ethylhexanal	-	-	8	10
3-heptanone	-	-	10	7
cyclohexanone	-	-	-	3
butyl ether	-	-	-	33

**Table 7 pathogens-12-00161-t007:** Indoor fungal strains isolated in Turku and used in liquid-based ^1^ or airborne ^2^ in vivo exposures of *C. elegans* nematodes.

Strain Code	Species/Genus	Origin of Strain	Medium
a/466	*Aspergillus fumigatus* ^1^	unknown	MEA
a/464	*Aspergillus niger* ^1,2^	unknown	MEA
Asp25	*Aspergillus ochraeus* ^1,2^	timber	MEA
a/459	*Chaetomium* sp. ^1,2^	chipboard/plastic carpet	MEA
b/548	*Geotrichium candidum* ^1,2^	negative control	MEA
a/462	*Paecilomyces variotii* ^1,2^	unknown	MEA
a/452	*Stachybotrys* sp. ^1^	toja board	MEA
a/453	*Stachybotrys* sp. ^1^	toja board	MEA
a/467	*Stachybotrys* sp. ^1,2^	timber	MEA
a/468	*Stachybotrys* sp. ^1^	mineral wool	MEA
a/470	*Stachybotrys* sp. ^1^	mineral wool	MEA
a/465	*Trichoderma* sp. ^1,2^	plastic paste	MEA
a/456	*Wallemia* sp. ^2^	mineral wool	DG18

**Table 8 pathogens-12-00161-t008:** Extracts ^1^ or exudates ^2^ of indoor fungal strains previously isolated in Helsinki or Espoo and used in in vivo, in vitro and ex vivo bioassays with *C. elegans*, porcine tubular kidney (PK-15) cells and boar sperm, respectively.

Strain Code and Reference	Species/Genus	Origin of Strain
POB8 [45]	*Agrostalagmus luteoalbus* ^1^	cork liner, Espoo
K20 [45]	*Aspergillus versicolor* ^1^	settled dust, Espoo
SL3 [45]	*Aspergillus versicolor* ^2^	fall out plate, Helsinki
MH1 [45]	*Chaetomium ^1^*	settled dust, Helsinki
OT7 = SzMC 24764 [61]	*Chaetomium cochliodes* ^1^	settled dust, Helsinki
ABCD [61]	*Chaetomium globosum* ^1^	settled dust, Helsinki
MH5 = SzMC 24456 [61]	*Chaetomium globosum* ^1^	settled dust, Espoo
TR [42]	*Penicillium sp*. ^1^	indoor air
RcP61 [45]	*Penicillium expansum* ^2^	cork liner, Espoo
SJ40 [42]	*Trichoderma citrinoviride* ^1^	settled dust, Espoo
THG = SzMC Thg [60]	*Trichoderma longibrachiatum* ^1^	insulation, Oulu
NJ14 [42]	*Trichoderma trixiae* ^2^	settled dust, Nivala

**Table 9 pathogens-12-00161-t009:** Biomass dispersals prepared in Helsinki from indoor fungal strains and used in in vivo, in vitro and ex vivo bioassays with *C. elegans*, baby hamster kidney (BHK-21) cells and boar sperm, respectively.

Strain Code	Species/Genus	Origin of Strain	Medium
A21	*Acremonium* sp.	surface dust, Vantaa	MEA
BA36	*Aspergillus* section *Aspergillus*	filter from exhaust air channel, Oulu	DG18
A4	*Aureobasidium pullulans*	surface dust, Vantaa	DG18
A5	*Aureobasidium pullulans*	surface dust, Vantaa	DG18
A6	*Aureobasidium pullulans*	surface dust, Vantaa	DG18
BA35	*Aureobasidium pullulans*	filter from exhaust air channel, Oulu	DG18
A3	*Exophiala* sp.	surface dust, Vantaa	MEA
BA32	*Exophiala* sp.	filter from exhaust air channel, Oulu	MEA
L6	*Paecilomyces* sp.	insulation, Jyväskylä	MEA
L21	*Paecilomyces* sp.	insulation, Jyväskylä	MEA
BA37	*Rhizomucor* sp.	filter from exhaust air channel, Oulu	MEA

## Data Availability

Not applicable.

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
