# Peer review of "Biomonitoring of Indoor Air Fungal or Chemical Toxins with Caenorhabditis elegans nematodes"

_pathogens, 2023, doi:10.3390/pathogens12020161_

Round 1

Reviewer 1 Report

In my opinion the manuscript mentioned below is scientifically sound and has been well written.    1. The authors have conducted research on using the nematode, Caenorhabditis elegans as a biomonitoring agent  for indoor air fungal and chemical toxins. This is currently an important research area and the issue addressed in this research is relevant, important and original. Nematodes have been commonly used as biomonitoring agents however, research on the use of them for biomonitoring air pollution is limited.    2. The authors have adopted standard scientific protocols and the results have been appropriately discussed.   3. The conclusions derived are valid and is warranted by the results obtained but can be further improved by been more specific and clearer.

Author Response

Thank you for the positive remarks on our manuscript. We have now added more figures and data to support our conclusions and have also modified the text to make our message more specific and clearer.

Reviewer 2 Report

In this manuscript, the authors examine the use of C. elegans as a biological sensor to detect toxins in indoor air. They utilize a transgenic stress reporter to assess toxicity to fungal and chemical toxins. The main conclusions of the manuscript are supported by the experimental data presented. However, there are several suggestions for improvement, including 

  • In the discussion section, the authors should discuss how the C. elegans model compares to previously tested in vitro and ex vivo methods. 
  • Please follow the standard c. elegans gene nomenclature system e.g., line 86, 209
  • Typos: line 88 - "also" please check and correct other typos too.
  • Please remove the borders for each figure panel.
  • In table 2, include a row for the mode of action of each toxin.
  • Fig 4A, provide representative pictures for each test condition and follow this approach for every figure.
  • Authors have used +++/++/+/- to denote the level of toxicity, and they should provide raw numbers (in a supp file) for each measurement.
  • Statistical significance should be noted for each comparison in each graph. 
  • The figure legend should include the statistical test type and mean/median values with errors. 

Author Response

Thank you for the positive remarks on our manuscript and for the suggestions that we have taken into account to improve the outcome, as detailed below in italics.

  • In the discussion section, the authors should discuss how the C. elegans model compares to previously tested in vitro and ex vivo methods.

In the discussion, the last paragraph has been dedicated to comparison of the methods, but it has now been extended.

  • Please follow the standard c. elegans gene nomenclature systemg., line 86, 209

We have used the European style nomenclature according to the article by Anbalagan et al. (2012), where the same C. elegans reporter strains have been used for biomonitoring purposes. 

  • Typos: line 88 - "also" please check and correct other typos too.

Our manuscript has now been checked and corrected by a native English-speaking group member.

  • Please remove the borders for each figure panel.

We have replaced all the figures and tables with updated versions. Borders have been removed from the figures, and we have also unified the styles and colors of the graphs. In addition, we have made small changes e.g. to make sure that the strain codes and their references are correct. 

  • In table 2, include a row for the mode of action of each toxin.

We have now provided a separate supplementary table with additional references for the known biological activities of the toxins listed in Table 2.

  • Fig 4A, provide representative pictures for each test condition and follow this approach for every figure.

We have now added additional representative examples of microscopic images to several figures.

  • Authors have used +++/++/+/- to denote the level of toxicity, and they should provide raw numbers (in a supp file) for each measurement.

As data for Tables 2 and 5 have been collected from multiple experiments, it is more feasible to summarize the average findings according to the scaling described in Table 1. However, for Tables 3 and 4, we have now added more detailed Supplementary data. In Tables 3 and 4, we have also separated the data obtained with the two distinct C. elegans reporter strains to emphasize the similarities in their responses.

  • Statistical significance should be noted for each comparison in each graph. 

We have now added statistical data to all bar charts.

  • The figure legend should include the statistical test type and mean/median values with errors.

We have now added information of statistical analyses into the graphs. Significant differences (P < 0.05) as compared to control samples were marked with asterisks; ns refers to no significance. Error bars represent standard deviations.